# Predictors of Psychological Well-Being and Quality of Life in Patients with Hypertension: A Longitudinal Study

**DOI:** 10.3390/healthcare12060621

**Published:** 2024-03-09

**Authors:** Maura Crepaldi, Jessica Giannì, Agostino Brugnera, Andrea Greco, Angelo Compare, Maria Luisa Rusconi, Barbara Poletti, Stefano Omboni, Giorgio Angelo Tasca, Gianfranco Parati

**Affiliations:** 1Department of Human and Social Sciences, University of Bergamo, 24129 Bergamo, Italy; jessica.gianni@unibg.it (J.G.); agostino.brugnera@unibg.it (A.B.); andrea.greco@unibg.it (A.G.); angelo.compare@unibg.it (A.C.); marialuisa.rusconi@unibg.it (M.L.R.); 2Department of Neurology, Laboratory of Neuroscience, IRCCS Istituto Auxologico Italiano, Piazzale Brescia 20, 20149 Milan, Italy; barbara.poletti@unimi.it; 3Department of Oncology and Hemato-Oncology, Università degli Studi di Milano, 20122 Milan, Italy; 4Clinical Research Unit, Italian Institute of Telemedicine, 21048 Solbiate Arno, Italy; stefano.omboni@iitelemed.org; 5Department of Cardiology, Sechenov First Moscow State Medical University, 119435 Moscow, Russia; 6School of Psychology, University of Ottawa, Ottawa, ON K1N 6N5, Canada; gtasca@uottawa.ca; 7Department of Cardiovascular, Neural and Metabolic Sciences, Istituto Auxologico Italiano, IRCCS, San Luca Hospital, 20149 Milan, Italy; gianfranco.parati@unimib.it; 8School of Medicine and Surgery, University of Milan-Bicocca, 20126 Milan, Italy

**Keywords:** hypertension, metabolic syndrome, psychological well-being, health-related quality of life

## Abstract

Previous research has highlighted the positive impact of greater health-related quality of life (Hr-QoL) and subjective well-being (SWB) on chronic diseases’ severity and progression. There is a paucity of studies investigating the long-term trajectories of these variables among hypertensive patients. The present study aims to investigate the relationships between psychological variables (Type A and D personality, locus of control—LoC, self-esteem, and trait anxiety) with SWB and Hr-QoL in patients with hypertension and comorbid metabolic syndrome. A total of 185 volunteer patients (130 males, 70.3%; mean age 54 ± 10.93) were enrolled. Patients filled out measures of Hr-QoL and SWB, LoC, and self-esteem at three time points—Type A and D behaviors and anxiety measures only at baseline. Analyses were run through two-level hierarchical mixed models with repeated measures (Level 1) nested within participants (Level 2), controlling for sociodemographic and clinical confounders. Neither Hr-QoL nor SWB changed over time. Patients with greater self-esteem and internal LoC (and lower external LoC) increased their SWB and Hr-QoL up to 1-year follow-up. A greater Type A behavior and trait anxiety at baseline predicted a longitudinal increase in most of the dependent variables. Results suggest that it could be useful to tailor interventions targeting specific variables to increase Hr-QoL and SWB among hypertensive patients.

## 1. Introduction

Hypertension is considered one of the diseases with significant costs to the health and economic systems of countries, as it is the leading cause of cardiovascular complications, such as stroke and myocardial infarction [1,2,3]. According to the Italian National Institute of Health, 31% of the Italian population aged between 35 and 74 are hypertensive, and 17% are borderline.

It is defined by a persistent high systolic (≥140 mmHg) and diastolic (≥90 mmHg) blood pressure (BP) [2]. It is a chronic medical condition that affects more than a quarter of the global population [4], and its prevalence increases with age. Most of its incidence across the age span seems to be due to systolic hypertension, while elevations of diastolic BP and isolated diastolic hypertension seem to be more common in younger rather than older patients [5]. European studies about hypertension during lifespan showed percentages of incidence included in a range between 2.2% and 13% in reference to adolescents, reporting that primary hypertension is the prevalent form during this stage of life [6]. However, the prevalence of hypertension increases with age, with a prevalence of 60% over the age of 60 years and 75% over the age of 75 years [5].

Hypertension requires comprehensive patient care because of its long-term impact on the body and patients’ quality of life [7]; if not controlled, it has severe but avoidable long-term consequences [3,8]. It is, globally, the most substantial modifiable risk factor for cardiovascular disease and related disabilities [9]. In addition to the symptoms of hypertension, in fact, many patients present additional risk factors such as metabolic syndrome (MetS). MetS refers to an agglomeration of cardiometabolic alterations and prediabetes conditions [10]; it is characterized by symptoms such as obesity, elevated blood pressure, insulin resistance, and dyslipidemia that are closely linked to metabolic malfunctioning and predispose individuals to develop Type 2 diabetes, cardiovascular disease, and renal events [11]. It follows that it would be of great importance and usefulness to limit risk factors for cardiovascular health, such as weight gain and physical inactivity [3]; physical inactivity, in this framework, seems to account for 5–13% of the risk for the development of hypertension [12]. Moreover, it is widely reported that regular physical activity helps the body reduce blood pressure [13].

Given the high incidence and prevalence rates of these conditions (hypertension and MetS) [9], a deeper understanding of their impact on patients’ health-related quality of life (Hr-QoL) and subjective well-being (SWB) seems to be useful. Hr-QoL is defined as the individual’s functioning in life concerning his perceived well-being in physical, mental, and social domains of health [14]. Literature has focused on the effects of hypertension on Hr-QoL, whose findings are rather heterogeneous due to the different components affected by the pathology condition; it has been hypothesized that such discrepancy may be related to methodological differences, including sampling criteria and use of different tools for the assessments [7]. Some studies showed worse levels of Hr-QoL in patients with hypertension than in normotensive individuals in mental and physical domains [15] as well as in general health [16], while others postulate that the relationship between hypertension and quality of life may be mediated by other variables, such as the awareness of the pathological condition [17]: so, more clarity on this field is still needed. Furthermore, the co-occurrence of several chronic diseases (e.g., metabolic syndrome or diabetes) in one person seems to amplify the results [18].

Well-being, on the other hand, refers to the subjective evaluation of one’s quality of life [19]. Studies focused on the impact of chronic diseases on SWB are widely reported in the literature, even if there are contrasting results regarding hypertension. In a study conducted by Stewart and colleagues [20], they found significant negative effects on SWB for many chronic cardiovascular conditions except for hypertension, which showed the least overall impact [20]. Conversely, a significant negative association between hypertension and well-being has been documented, highlighting lower levels of SWB in this pathological condition [21]⁠. Schaare and colleagues [22] recently tried to clarify controversies about the relationship between blood pressure (BP), hypertension, and well-being. They analyzed and monitored normotensive subjects (some of whom developed hypertension over time—up to 5 and 10 years from the enrolment) and patients with a diagnosis of hypertension (27% of the sample). They found a positive association between systolic BP and well-being in normotensive patients both on a cross-sectional level and longitudinally, but the same cannot be said for hypertensive patients, who developed worse well-being.

In the literature on hypertension, the role of psychological factors, such as self-esteem and locus of control (LoC), have also been analyzed. In particular, they have been considered the main factors influencing health behavior [23,24]. Self-esteem refers to the individual subjective evaluation of one’s worth as a person [25]. The relationship between self-esteem and SWB has been long investigated in the literature on chronic diseases; recently, Islam and Ara [26] highlighted a significant positive relationship between self-esteem and indexes of mental and physical well-being in different chronic conditions. Moreover, lower levels of self-esteem, together with physical inactivity and lower optimism, have been regarded as heavy risk factors for cardiovascular health [27]. LoC, on the other hand, refers to the individual’s belief about control and power they can have over events in their life [28]. It is composed of internal Loc, which refers to the individual’s perception that the event is contingent upon his own behavior, and external Loc, which corresponds to the individual’s perception that the event is not entirely contingent upon his action [28]. Evidence showed that internal LoC implies an increased patient belief that they have some control over their health, believing that their attitudes and actions can positively impact their well-being [20]. Furthermore, an external LoC, together with irrational health beliefs, has recently been treated as a risk factor for the development of hypertension, estimating that health-related irrational beliefs make hypertensive patients consider themselves as less likely to be at risk than those around them and thus have less health-focused behaviors [29].

Literature has also focused on the link between anxiety and cardiovascular conditions, given the negative impact of high trait anxiety in the pathophysiology of hypertension in MetS [10]. Trait anxiety refers to the individual differences in anxiety-proneness as a personality trait, so persons with higher trait anxiety are more strongly disposed to manifest anxiety state [30]. In this field, many studies reported high levels of trait anxiety in patients with hypertension [31]; furthermore, trait anxiety has been associated with poorer Hr-QoL [31] and lower levels of SWB [32] in patients with hypertension compared to the general population. However, the causal effect between these pathological conditions and the well-being sphere remains unclear and needs further clarification [4].

Finally, hypertension’s etiology and prognosis have been linked to personality traits (i.e., Type A and Type D), which seem to significantly affect patients’ Hr-QoL [33,34]. Type A behavior corresponds to a profile characterized by high competition, ambition, and motivation to achieve, as well as impatience, aggressiveness, social hostility, and vulnerability to stress [27]. Recently, the presence of Type A traits in patients affected by cardiovascular diseases has been associated with an impairment of Hr-QoL over time [35]. Type D, on the other hand, characterizes those individuals who experience increased negative emotions and tend to inhibit themselves in social interactions [27]. Studies indicated this last pattern as the most significant predictor of worse quality of life in hypertensive patients [36], and it has been associated with low levels of SWB among cardiovascular patients for decades [37].

Even if Hr-QoL and SWB seem to be compromised in patients with hypertension, there is a limited body of research examining the determinants of these two variables among these patients [38]. For this reason, the present work aims to (a) investigate the trajectories of Hr-QoL and SWB over time to better clarify their trend over patients with hypertension and MetS and (b) investigate psychological and personality variables (LoC, self-esteem, trait anxiety, Type A and Type D traits) significantly affecting the longitudinal changes over time in Hr-QoL and SWB. We expect that patients will perceive better well-being and quality of life due to the monitoring of their BP, as we can see in previous work [39], and that the above-mentioned psychological and personality traits will significantly predict this trend; in particular, we expect an impact of self-esteem and internal LoC [24,26], as well as an influence of Type A, Type D behaviors [35,36,37] and trait anxiety [31,32] on both mean outcomes (Hr-QoL and SWB) according to previous studies.

## 2. Materials and Methods

### 2.1. Participants

A total of 185 patients with hypertension and co-occurring metabolic syndrome volunteered for this study. They all showed signs of metabolic syndrome and were at an increased risk of cardiovascular events in the short term. For this reason, adequate blood pressure control was a clinically relevant issue for these patients.

All patients included in the sample had to meet the following inclusion criteria: (i) age between 18 and 74 years; (ii) untreated or treated arterial hypertension according to both offices (systolic BP ≥ 140 and/or diastolic BP ≥ 90 mmHg) and day-time blood pressure (systolic BP ≥ 135 and/or diastolic BP ≥ 85 mmHg) and (iii) presence of metabolic syndrome according to Adult Treatment Panel (ATP) III criteria [8]. Main exclusion criteria were: (i) secondary hypertension; (ii) renal or liver impairment; (iii) severe autoimmune diseases; (iv) neoplasia; (v) atrial fibrillation, frequent ectopic beats or severe conduction disturbances; (vi) an arm circumference >32 cm or <22 cm; (vii) any other condition that could have compromised the patient’s participation in the study, including illiteracy and diseases preventing the completion of questionnaires (for example, myopia); (viii) patients not able to follow study procedures.

### 2.2. Procedure

The study is an Italian multicenter trial with an open-label, parallel-group, randomized, controlled design. It involves 12 hypertension centers: seven located in northern Italy, two in central Italy, and three in southern Italy.

Patients were first screened for inclusion and exclusion criteria. During the first visit, weight, height, and circumferences were measured by trained professionals, and BMI (Body mass index) was calculated considering the ratio of weight to the square of height (weight (kg)/height*height (m^2^)). Second, patients were subjected to a week of run-in and then randomized to two different arms (ratio 2:1) using a blocked randomization procedure: (i) an intervention group involved in treatment as usual (TAU, which included pharmacotherapy) matched to the automated teletransmission of BP values taken at home to the investigator; the transmission took place every month for the first 24 weeks and in conjunction with regular doctor’s visit every three months in the second 24 weeks (see the protocol for further details [40]), and (ii) a control group exclusively treated with regular office visits every three months (TAU). All patients underwent 24-h ambulatory BP monitoring at recruitment (baseline), at 24 and 48 weeks of follow-up, while their office BP was measured at recruitment (baseline) and one week, 4, 12, 24, 36, and 48 weeks follow-up for a total of 7 different time points. Furthermore, patients’ antihypertensive drug treatment was adjusted to achieve office BP normalization (<140/90 mmHg) on the next visit.

During three visits (1, 24, and 48 weeks follow-up), patients completed a battery of psychological measures, which included a measurement of well-being, a measure of quality of life, and locus of control and self-esteem questionnaires (see the next paragraph for details). Further dimensions have been taken into account at Visit 2 (1 week after the recruitment) to investigate psychological and personality traits of hypertensive patients with comorbid metabolic syndrome (trait anxiety and personality Type A and Type D).

The RCT was conducted according to Good Clinical Practice guidelines and the Declaration of Helsinki, approved by the Ethics Committees of the centers involved and registered at ClinicalTrials.gov (registration number: NCT01541566). Participants provided written informed consent before their voluntary enrolment.

### 2.3. Measures

#### 2.3.1. Main Outcomes

The Italian version of the SF-36 Health Survey Questionnaire [41] is a 36-item self-report multidimensional scale that measures both physical (PCS) and mental components (MCS) of health status. These two components are divided into eight scales, namely physical functioning, physical role, bodily pain, general health perceptions (which form the PCS subscale), and vitality, social functioning, emotional role, and mental health (which form the MCS subscale). Total component scores range from 0 to 100, with higher scores indicating greater perceived physical and mental health quality of life. A previous study by Apolone and Mosconi (1998) demonstrated the good psychometric properties of the scale [42]. In the current study, Cronbach’s Alphas across all time points were good to excellent (PCS, α = 0.77; MCS, α = 0.81).

The Italian version of the Psychosocial General Well-Being Index (PGWB; [43]) is a 22-item self-report questionnaire that evaluates psychological and general well-being during a 4-week period. Each item is rated on a 6-point Likert scale (0 to 5). Total scores range from 0 to 110, with higher scores indicating greater well-being. A previous study by Grossi and colleagues (2006) demonstrated the good psychometric properties of the scale [44]. In the current study, Cronbach’s Alpha across all time points was excellent (α = 0.95).

#### 2.3.2. Time-Varying Psychological Predictors

The Italian Version of the Locus of Control of Behavior (LCB; [45,46]) is a 17-item self-report questionnaire that measures the tendency to attribute the causes of behavior to internal or external conditions. Each item is rated on a 6-point Likert scale (0 to 5). Greater subscale scores indicate a higher internal (evaluated through 7 items) or external (evaluated through 10 items) LoC. A previous study by Wall and colleagues (1989) demonstrated the good psychometric properties of the scale [47]. In the current study, Cronbach’s Alpha across all time points was good (internal LoC, α = 0.74; external LoC, α = 0.78).

The Italian version of the Rosenberg Self-Esteem Scale (RSES; [48]) is a 10-item self-report questionnaire that measures the degree of self-esteem. Each item is rated on a 4-point Likert scale ranging from 1 (“strongly agree”) to 4 (“strongly disagree”). Total scores range from 1 to 40, with higher scores indicating greater self-esteem. A previous study by Alessandri and colleagues (2015) demonstrated the good psychometric properties of the scale [49]. In the current study, Cronbach’s Alpha across all time points was excellent (α = 0.84).

#### 2.3.3. Psychological Predictors Measured Only at Baseline

The Italian version of the Trait Anxiety Inventory (STAI-Y; [50]) is a 20-item self-report questionnaire that assesses the degree of trait anxiety. Each item is rated on a 4-point Likert scale with a range from 1 to 4. The total score goes from 20 to 80. A previous study by Ilardi and colleagues (2021) demonstrated the good psychometric properties of the scale [51]. In the current study, Cronbach’s Alpha at baseline was excellent (α = 0.89).

The Type D Scale-14 (DS-14; [52]) is a 14-item self-report questionnaire that evaluates the presence of personality traits defined as being distressed. Each item is rated on a 5-point Likert scale ranging from 0 to 4. The total score goes from 0 to 56, with higher scores indicating a more distressed personality. A previous study by Gremigni and Sommaruga (2005) demonstrated the good psychometric properties of the scale [53]. In the current study, Cronbach’s Alpha at baseline was excellent (α = 0.89).

Type A behavior pattern (TABP) was assessed through the Italian version of the semi-structured interview for Diagnostic Criteria for Psychosomatic Research (DCPR) [54]. Questions concern the past 6 or 12 months and include 9 items (A) that require a yes/no dichotomous answer, each representing 9 different characteristics of Type A personality. One last item (B) requires the same type of answer, and it is related to physical manifestations of the previous characteristics. Responses to the 9 questions can be summed, leading to a total score, with higher scores suggesting a worse Type A behavior. In the current study, Cronbach’s Alpha at baseline was good (α = 0.70).

#### 2.3.4. Blood Pressure

Office BP measurement was obtained thanks to an automatic device. After 5 min of sitting, automated BP was measured twice at a 2-min interval, and the performances obtained were averaged [40].

### 2.4. Statistical Analysis

We examined the longitudinal changes in well-being and quality of life among patients with hypertension and comorbid metabolic syndrome through two-level Hierarchical Linear Models (HLMs), with repeated measures (Level 1) nested within participants (Level 2).

At Level 2, HLM analyses were adjusted for several grand-mean-centered control variables, including demographic and clinical variables, namely randomization group (control vs. intervention), sex, any previous cardiovascular disease (not present vs. present), end organ damage, and number of antihypertensive drugs taken. In addition, the total number of risk factors for hypertension was added as a group-centered time-varying covariate (TVC) at Level 1 of the HLM models. Based on the criteria provided by the European Society of Hypertension and by the European Society of Cardiology [55], each of the following was considered a risk factor: age > 54 (for males) or >64 (for females), smoking status, dyslipidemia, diabetes, BMI ≥ 30, Waistline ≥ 102 (for males) or ≥88 (for females), and family history of hypertension. The number of risk factors (0 to 7) led to the final composite score.

As regards the psychological variables, those that were evaluated at several time points (namely quality of life, locus of control, and self-esteem) were entered as TVCs at Level 1 and centered around the individual’s mean, while the psychological variables examined only at baseline (namely trait anxiety, Type D and Type A) were entered at Level 2 of our models and centered around the grand mean [56]. The multilevel model is reported in the Appendix A.

As for the effect sizes, we assessed and reported pseudo-R^2^, a measure of the proportion of within-person variance accounted for by adding the linear time parameter [56], the magnitude of which was interpreted according to guidelines [57].

Analyses were performed using the Statistical Package for Social Sciences (SPSS) version 28.0 and Hierarchical Linear Models (HLM) Professional version 8.2. HLM’s model parameters were estimated using the restricted maximum likelihood method with robust standard errors. All statistical tests were two-tailed, and a *p* ≤ 0.05 was considered statistically significant.

## 3. Results

### 3.1. Preliminary Analyses

Of the total sample, all composed by Caucasian, 130 were males (70.3%) with a mean age of 54 years (SD = 10.93); their mean BMI was 30.6 kg/m^2^ (SD = 4.2) and their mean waist circumference was 109.4 cm (SD = 10.3) (see Table 1; for further details) [39].

All variables were normally distributed (i.e., all skewness and kurtosis were ≤1 [58]) except for the number of antihypertensive drugs taken, which was positively skewed. This variable was log-transformed to correct the violation of the assumption of normality. Furthermore, we found no univariate outliers among the data (i.e., all the z-transformed scores were ≤2 [58]). All measures had a good reliability (see Table 2).

The unconditional models without covariates demonstrated that patients experienced significant, longitudinal increases in well-being (as measured by the PGWB) and in the mental components of quality of life (as measured by the SF-36 MCS subscale), from the recruitment in the study up to 48 weeks later (1 year of follow-up), with large effects. No significant changes over time in the physical components of quality of life (as measured by the SF-36 PCS subscale) were observed, even if this effect was large. Means, standard deviations at each time point, and detailed results from these analyses are reported in Table 2 and Table 3, respectively.

### 3.2. Main Analysis

First, the longitudinal changes in all dependent variables (i.e., well-being and the physical and psychological components of QoL), while controlling for all the covariates (including sociodemographic, clinical, and psychological ones), were no more significant.

As for PGWB, patients with greater levels of trait anxiety reported lower well-being at baseline than those with lower trait anxiety. Furthermore, those with both greater TABP or trait anxiety at baseline reported a significant increase in well-being up to 48 weeks follow-up. Finally, patients with greater internal locus of control and self-esteem and lower external LoC experienced higher levels of well-being during all time points.

As for SF-36 PCS, those who reported lower levels of trait anxiety at baseline experienced a significant increase in the physical components of QoL over time. Further, a greater internal locus of control was positively associated with the dependent variable during all time points.

As for SF-36 MCS, lower trait anxiety was associated with greater mental health components of QoL at baseline. However, those with higher levels of trait anxiety and TABP reported a significant increase in the dependent variable over time. Finally, greater self-esteem and a lower external locus of control were significantly associated with heightened mental health components of QoL during all time points. Detailed results (including B coefficients, degrees of freedom, *t*- and *p*-values) are reported in the Appendix A.

## 4. Discussion

The present study shows the results of a longitudinal design aimed at investigating the trajectories of SWB and Hr-QoL and their predictors over time in patients with hypertension and comorbid metabolic syndrome. In particular, the effects of each predictor (i.e., LoC, self-esteem, anxiety, and personality traits) on the longitudinal changes in the dependent variables were discussed. Results were discussed considering the unbalanced composition of the sample with respect to gender, which is in line with the literature, reporting higher rates of hypertension among males than females [59].

### 4.1. Trajectory of SWB and Hr-QoL over Time

As for the uncontrolled analyses on all the dependent variables, we found a significant, longitudinal increase in the psychological components of Hr-QoL and SWB from the recruitment in the study up to 48 weeks into treatment. Interestingly, the physical components of Hr-QoL did not significantly change over time. We suppose that—once recruited in the study—our patients experienced an overall reduction in blood pressure [39] and better care than usual (with approximately seven appointments with a physician in less than 48 weeks), leading to an increase in quality of life and well-being over time. It could be argued that self-reported physical components of QoL [41] are not affected by hypertension in the short term (i.e., within 1 year) if a patient both adheres to pharmacological treatment and pursues lifestyle changes [3,8]: this may explain why our patients did not experience a change in this variable over time.

Finally, we found that all longitudinal changes became non-significant, suggesting that other variables may better account for the observed increase up to 48 weeks follow-up in some of the dependent variables.

To explore this trend and to identify the predictors that most affect the changes in SWB and Hr-QoL over time, psychological and personality traits have been considered, and results are presented in the paragraphs below.

### 4.2. Locus of Control and Self-Esteem

According to the previous literature and with our hypothesis [24,48], we found that internal LoC and self-esteem were significantly and positively associated with SWB and Hr-QoL over time, while external LoC was negatively associated with SWB and psychological QoL up to 48-weeks follow-up.

This may be due to a high internal LoC, which may lead to an increased focus on behaviors that compete to achieve better health and, thus, well-being. Indeed, the literature argues that individuals with a high internal LoC tend to be motivated to take voluntary actions to promote their own health and disease management [24,49]. In contrast, we found an inverse trend for external LoC, being negatively associated with a better QoL over time. It is known from the literature that high external LoC implies a lack of responsibility in disease management, and subjects tend to be less proactive in following treatment guidelines or health self-management (e.g., blood pressure measurement and therapy adherence [60]).

Together with LoC, self-esteem is an essential factor in influencing general health conditions and mental health [61,62] in cardiovascular patients. According to previous literature and our hypothesis, in our study, patients with high self-esteem showed greater SWB up to 48 weeks after recruitment, indicating that a positive self-assessment attitude and the feeling of being able to cope with the disease are linked to better well-being and psychological QoL. Indeed, self-esteem seems to play a crucial role in mediating relationships between mental-sphere variables (e.g., life satisfaction and well-being [7]) and general health-sphere (e.g., stress and cardiovascular health [63]). Furthermore, studies have shown that high levels of self-esteem can be associated with both a decrease in the incidence of coronary heart disease [64] and improved Hr-QoL in patients with heart failure [65], demonstrating its crucial role in patients’ overall health-sphere [62].

### 4.3. Anxiety and Personality Traits

According to our hypothesis, we found negative correlations between trait anxiety and both psychological measures at baseline (PGWB and SF-36—MCS subscale), indicating that anxious hypertensive patients feel worse in their psychological sphere. Longitudinally, we found an interesting trend that goes far from our hypothesis: patients with higher trait anxiety and TABP traits showed greater well-being and psychological QoL over time. However, it is crucial to consider that initial poorer well-being and QoL are observed in anxious patients, and such levels increase a little over time, whereas patients who are not anxious tend to feel better, and their psychological health tends to stabilize over time (as shown in Appendix A).

As for the physical domain (SF-36—PCS subscale), we found that lower anxiety was associated with greater physical QoL over time. No other significant relations were found with respect to Type D personality traits.

Regarding anxiety, it is crucial to underline that data on its relationship with hypertension still appears inconsistent [66,67]. According to previous evidence [68,69], different anxiety symptoms (e.g., generalized anxiety disorder) had already shown contrasting effects on cardiovascular diseases (CVD) over time, suggesting that the anxiety disorder could work as a “protective” variable. These studies hypothesized that people who experience more anxiety tend to behave better in the health dimension, addressing their health needs more regularly and conscientiously and using more healthcare services; moreover, it has been observed that anxiety disorders might improve compliance activities [70]. Blumenthal and colleagues [71] extended this protective role to trait anxiety, suggesting that highly anxious patients behave in a way that helps them overcome cardiovascular disease outcomes. Focusing on hypertension, Bardage and Isacson [66] found that this pathologic condition mainly influences the general health perception (reflecting both mental and physical health). This could be the mechanism that brings anxious patients to behave better to preserve their health; it also may explain why, in our results, these patients increased their well-being and psychological QoL over time. The physical domain is equally influenced by these behaviors, thus confirming that hypertensive patients who adhere to pharmacological treatment and implement lifestyle changes also have better physical QoL [3,8]. However, physical QoL showed a different trend over time because it is associated with poorer trait anxiety: this could be due to the negative consequences that anxiety brings to our body and our health-sphere as a whole [10].

In conjunction with the anxiety dimension, individual predisposition plays a crucial role in implementing healthy behaviors [35]. According to previous evidence [72], the individual ability to adjust to mental and health-related demands and to cope emotionally with the disease causes patients to develop adaptive versus maladaptive disease-promoting behavioral patterns. TABP, reflecting a marked vulnerability to stress but also an ambitious and competitive profile [27], may explain our results: they reflect the association with altered levels of distress experienced by patients in strict relation to their condition, thus amplifying motivation to act so they behave healthier.

## 5. Strengths of the Study and Limitations

The presented study shows promising results regarding the trend of SWB and Hr-QoL in hypertensive patients over time, in particular concerning the role of different psychological predictors on this trend, trying to clarify contrasting results from the literature. Therefore, following the patients for longer periods of time would be worthwhile.

Despite this, the study has some limitations. First, we considered an Italian sample; it might be interesting to propose the study in other countries to explore regional variations and verify the generalizability of the results. Moreover, we cannot exclude the fact that the results could have shown a different trend if the follow-up time had been longer. Finally, we decided to focus on specific psychological predictors; exploring the 24 parameters of ambulatory BP monitoring in different personality types and more psychological factors (e.g., anger, hostility, stress, depression) in future studies might be valuable.

## 6. Conclusions

The present study confirmed the importance of studying the predictors of SWB and Hr-QoL and their development over time to clarify the conflicting results in the literature on hypertensive patients.

The study showed that high self-esteem and internal LoC seem to protect patients’ well-being, so it may be useful to propose interventions improving self-esteem and internal LoC skills, as patients feel in control of their health and behavior, which also affects their well-being. Furthermore, our results shed light on the role of trait anxiety and personality traits (Type A and D), showing that anxious and competitive, motivated and ambitious patients (characteristic features of the Type A profile) tend to be more focused on their health and act strictly to control and monitor their condition.

Moreover, these results could have important implications in clinical practice, such as the development of personalized intervention aimed at improving well-being and Hr-QoL in patients with hypertension and metabolic syndrome in comorbidity, as well as disease management and treatment adherence by isolating and considering the individual role of these variables over time, thus promoting interventions based on the individual profile of patients.

## Figures and Tables

**Table 1 healthcare-12-00621-t001:** Sociodemographic and Clinical Characteristics of Participants (N = 185).

Variables	Age	Body Mass Index (BMI)	Waistline	Number of Target Organs Affected	Number of Antihypertensive Drugs Taken	History of Cardiovascular Disease
M (SD)	54 (10.9)	30.6 (4.2) kg/m^2^	109.4 (10.3) cm	median 2	1.95 (1.11)	
n (%)						10 (5.4)

**Table 2 healthcare-12-00621-t002:** Cronbach’s Alpha for all measures. Means and Standard Deviations for all Psychological Variables at all time points (Baseline, 24- and 48-weeks of follow-up) in the sample of 185 patients with hypertension and co-occurring metabolic syndrome.

	Baseline			24-Weeks		48-Weeks	
Variable(s)	Cronbach’s Alpha	N	Mean (SD)	N	Mean (SD)	N	Mean (SD)
PGWB	0.95	181	73.94 (17.17)	170	75.09 (17.65)	171	76.74 (18.31)
SF-36 PCS	0.77	177	48.23 (8.52)	167	47.38 (9.30)	169	47.57 (8.76)
SF-36 MCS	0.81	177	43.23 (10.89)	167	45.30 (10.24)	169	46.13 (10.45)
LoC Internal	0.74	174	23.76 (5.94)	166	23.55 (6.42)	167	24.14 (6.49)
LoC External	0.78	174	19.63 (8.90)	166	19.20 (9.16)	167	19.07 (9.23)
RSES	0.84	178	30.47 (4.95)	166	30.41 (5.31)	169	31.18 (5.11)
STAI-Y Trait	0.89	173	39.47 (10.68)	\	\	\	\
DS-14	0.89	175	20.84 (11.75)	\	\	\	\
TAB	0.70	174	3.47 (2.21)	\	\	\	\

Note: PGWB = Psychosocial General Well-Being Inventory; SF-36 = SF-36 Health Survey Questionnaire (PCS = Physical Component Scale; MCS = Mental Component Scale); RSES = Rosenberg Self-Esteem Scale; LoC = Locus of Control of Behavior; STAI-Y = Trait Anxiety Inventory; DS-14 = Type D Scale 14; TABP = Type A behavior pattern.

**Table 3 healthcare-12-00621-t003:** Fixed effects for the uncontrolled longitudinal changes (from baseline up to 48 weeks of follow-up) in well-being and quality of life in the sample of 185 patients with hypertension and co-occurring metabolic syndrome.

Variable(s)	B	SE	t-Value	df	*p*-Value	R^2^
PGWB	0.055	0.028	2.006	183	0.046	0.27
SF-36 PCS	−0.021	0.015	−1.415	182	0.159	0.21
SF-36 MCS	0.059	0.019	3.148	182	0.002	0.23

Note. SE = Standard Error; df = degrees of freedom; PGWB = Psychosocial General Well-Being Inventory; SF-36 = SF-36 Health Survey Questionnaire (PCS = Physical Component Scale; MCS = Mental Component Scale). R^2^ refers to pseudo-R^2^, indicating the proportion of within-person variance accounted for by adding the “Time” parameter to the model.

## Data Availability

The data that support the findings of this study are available from the corresponding author, [M.C.], upon reasonable request.

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
