# Peer review of "Predictors of Psychological Well-Being and Quality of Life in Patients with Hypertension: A Longitudinal Study"

_healthcare, 2024, doi:10.3390/healthcare12060621_

Round 1

Reviewer 1 Report

Comments and Suggestions for Authors

The study aims to investigate the trajectories of HrQoL and SWB over time in individuals with hypertension and Metabolic Syndrome, while also exploring the impact of psychological and personality variables on these longitudinal changes, anticipating improvements in well-being under blood pressure monitoring.

The manuscript is clear and well-structured, presenting relevant and innovative topics to the field. The introduction is adequately elaborated and the materials and methods clearly presented. The results are displayed logically and in an orderly manner. However, some suggestions are made.

Line 50: "Given the high incidence and prevalence rates of these conditions". The introduction would be more complete if such data were provided.

Lines 136-139: I suggest that the authors move this sample descriptive paragraph and Table 1 to the beginning of the Results section.

Lines 156-182: Authors are requested to provide in this section the necessary information for BMI: How was it calculated (formula and reference)? Were weight, height, and circumferences measured by trained professionals or self-reported by participants?

Tables are appropriate, properly show the data, and are easy to interpret and understand. Discussions and conclusions are consistent with the evidence and arguments presented.

Author Response

Dear Reviewer,

Thank you for the suggestions; below the answer to your requests.

Line 50: "Given the high incidence and prevalence rates of these conditions". The introduction would be more complete if such data were provided.

Thank you for your suggestion. We have added references to complete the introduction better.

Lines 136-139: I suggest that the authors move this sample descriptive paragraph and Table 1 to the beginning of the Results section.

Thank you for your suggestion. We moved the description of the sample and Table 1 to the beginning of the Results section, as requested.

Lines 156-182: Authors are requested to provide in this section the necessary information for BMI: How was it calculated (formula and reference)? Were weight, height, and circumferences measured by trained professionals or self-reported by participants?

We gave additional information about how BMI was calculated (lines 174-178)

Reviewer 2 Report

Comments and Suggestions for Authors

Dear Authors,

I've read with great interest the manuscript titled “Predictors of psychological well-being and quality of life in  patients with hypertension: a longitudinal study” with a really interesting aim.

To the best of my knowledge, it is well-structured, the methodology seems adequate, results are nicely presented and the conclusions are supported by the results obtained.

Authors have described an important topic which is rarely used in everyday clinical setting in this multicentric study from Italy. 

Although it is not the goal of the current study, it would be interesting to investigate whether there are differences in the 24 ABPM parameters in different personality types. To find out if there are regional variations, it would also be fantastic to conduct an international study in Europe.

All things considered, I am recommending the publication of your manuscript.  

Best regards, Peer Reviewer

Author Response

Dear Reviewer,

Thank you for the suggestions; below the answer to your requests.

Although it is not the goal of the current study, it would be interesting to investigate whether there are differences in the 24 ABPM parameters in different personality types. To find out if there are regional variations, it would also be fantastic to conduct an international study in Europe.

Thank you for this recommendation. We added this information in the “Limitations” section as future perspectives

Reviewer 3 Report

Comments and Suggestions for Authors

Abstract : Please add information on the statistical analysis method used in this study in abstract.

Introduction : The average age of this study is 54, and most of them are in their 40s and 60s. Hypertension often develops after 40s, but please further supplement the analysis of previous studies on hypertension in adolescents in their 20s and 30s.

2.3

The reliability value of the questionnaire was written only in writing, and please add that the reliability of the questionnaire was secured for each question by using the table in which the Cronba alpha value for each question was presented. Also, please add how you conducted the validity verification of the questionnaire.

 You presented the discussion from the perspective of LoC, self-estem, anxiety and personalities. In addition, I would like you to add what physical activity means for health and why physical activity is necessary for patients experiencing high blood pressure and metabolic syndrome.

 5. In addition to the limitations, please present the strengths and characteristics of this study.

Author Response

Dear Reviewer,

Thank you for the suggestions; below the answer to your requests.

Abstract : Please add information on the statistical analysis method used in this study in abstract.

Thank you for your suggestion. We included additional information in the abstract about the statistical analysis method used.

Introduction : The average age of this study is 54, and most of them are in their 40s and 60s. Hypertension often develops after 40s, but please further supplement the analysis of previous studies on hypertension in adolescents in their 20s and 30s.

Thank you for the feedback, we included additional information about the incidence of hypertension during lifespan  (lines 47-54)

The reliability value of the questionnaire was written only in writing, and please add that the reliability of the questionnaire was secured for each question by using the table in which the Cronba alpha value for each question was presented. Also, please add how you conducted the validity verification  of the questionnaire.

Thank you for your suggestion. We added Cronbach Alpha values in Table 2 and the references about the verification of validity under each questionnaire used.

You presented the discussion from the perspective of LoC, self-estem, anxiety and personalities. In addition, I would like you to add what physical activity means for health and why physical activity is necessary for patients experiencing high blood pressure and metabolic syndrome.

Thank you; we added your suggestion in the first section of the article (Introduction, lines 64-68). The discussion of our results is focused on the physical component of quality of life and we didn’t consider the amount of physical activity of our sample.

In addition to the limitations, please present the strengths and characteristics of this study.

Thank you for the suggestion. We have reworded the paragraph on limitations and added the strengths as well.

Reviewer 4 Report

Comments and Suggestions for Authors

The main research question is to verify if there is a relationship between psychological variables, subjective well-being and health-related quality of life in hypertensive patients with comorbid metabolic syndrome.

The topic is original and addresses an existing gap in this field.

There are more men than women in the sample  (70.3%), with a significant difference. This fact could have influenced the results, and yet it is not mentioned as a limitation.

Bibliographic references are appropriate.

Author Response

Dear Reviewer,

Thank you for the suggestions; below the answer to your requests.

There are more men than women in the sample  (70.3%), with a significant difference. This fact could have influenced the results, and yet it is not mentioned as a limitation.

Bibliographic references are appropriate.

Thank you for emphasizing this point. We have added more information on the prevalence of males in the "discussion" section (lines 360-363); the fact that there are more males than females in the sample aligns with the literature that reports higher rates of hypertension among males in the Italian population.